



**MODIFICATIONS TO KOZENY-CARMAN MODEL To**
**ENHANCE PETROPHYSICAL RELATIONSHIPS**
Amir Maher Sayed Lala
Geophysics Department, Ain Shams University
**e-mail: amir77_lala@yahoo.com**
**Affiliation:** Geophysics Department, Fac. of Science, Ain Shams University





**MODIFICATIONS TO KOZENY-CARMAN EQUATION To**
**ENHANCE PETROPHYSICAL RELATIONSHIPS**
Amir M. S. Lala
Geophysics Department, Ain Shams University

27                              **Abstract**

The most commonly used relationship relates permeability to porosity, grain

size, and tortuosity is Kozeny-Carman formalism. When it is used to estimate the
permeability behavior versus porosity, the other two parameters (the grain size and
tortuosity) are usually kept constant. Here, we investigate the deficiency of the
Kozeny-Carman assumption and offer alternative derived equations for the Kozeny-
Carman equation, including equations where the grain size is replaced with the pore
size and with varying tortuosity. We also introduced relationships for the permeability
of shaly sand reservoir that answer the approximately linear permeability decreases in
the log-linear permeability-porosity relationships in datasets from different locations.

**Introduction**

Darcy's law (e.g., Mavko et al., 2009) states that, the volume flux of viscous

fluid Q (volume per time unit, e.g., m$^3$/s) through a sample of porous material is
proportional to the cross-sectional area A and the pressure difference $\Delta$P applied to
the sample's opposite faces, and inversely proportional to the sample length L and the
fluid's dynamic viscosity $\mu$, as shown as follows :
$Q = -k\frac{A}{\mu}\frac{\Delta P}{L}$ … … … … … … … … … … … … … … … … … … … … … … … … … … … … … … … .(1)

The proportionality constant $k$ is called the absolute permeability. The main

assumption of Darcy's law is that, $k$ does not depend on the fluid viscosity $\mu$ or
pressure difference $\Delta P$. All inputs in equation 1 have to have consistent units,



meaning that if length is in m, pressure has to be in Pa and viscosity in Pa s. The most
commonly used viscosity unit is cPs = $10^{-3}$ Pa s. It follows from Equation 1 that the
units of k are length squared, e.g., $m^2$. The most common permeability units used in
the industry are Darcy (D) and/or milliDarcy (mD): $1D = 10^{-12}$ $m^2$ and 1 mD = $10^{-15}$
$m^2$.
The Kozeny-Carman (KC) formalism (e.g., Mavko et al., 2009) assumes that a
porous solid can be represented as a solid block permeated by parallel cylindrical
pores (pipes) whose axes may be at an angle to the direction of the pressure gradient,
so that the length of an individual pipe is larger than that of the block. To relate
permeability to porosity in such idealized porous solid we need to find how the
volume flux $Q$ relates to the pressure gradient $\Delta P$. The solution is based on the
assumption that each cylindrical pipe is circular, with radius $r$. The Navier-Stokes
equations governing laminar viscous flow through a circular pipe of radius $r$ provide
the following expression for the volume flux $q$ through an individual pipe:

$$q = -\frac{\pi r^4}{8\mu}\frac{\Delta P}{l} \dots\dots\dots\dots\dots\dots\dots\dots\dots\dots\dots\dots\dots\dots\dots\dots\dots (2)$$

where: $l$ is the length of the pipe.
Our derivation starts from the Kozeny-Carman equation by assuming that a
rock includes porosity of pipe shape. Permeability of this rock is expressed by its
porosity $\varphi$ and the specific surface area $S$ to the radius of an individual pipe, its
length, and the number of the pipes, and using Equation 1, we get:
$$k = \frac{1}{2}\frac{\varphi^3}{S^2\tau^2} \dots\dots\dots\dots\dots\dots\dots\dots\dots\dots\dots\dots\dots\dots\dots\dots\dots\dots. (3)$$





where: $S$ is defined as the ratio of the total pore surface to the total volume of the
porous sample and the tortuosity $\tau$ is simply $l / L$ , defined as the ratio of the length of
the fluid path to that of the sample. Porosity can be evaluated in the laboratory or
obtained from porosity logs. The specific surface area is much more difficult to
measure or infer. One other parameter that can be detremined in the laboratory is the
average grain size (diameter) d. This is why it is possible to conduct relationship
between $k$ and d. So modified Kozeny– Carman equation is needed if a non-fractal
spherical grain packing model is assumed (yielding a constant tortuosity) and the
effective pore radius is substituted by a term involving the specific surface expressed
by the grain radius and the porosity. This operation is inconsistent with the KC
formalism but it is useful. Assume that the number of these grains is n, their volume is
$n\pi d^3 / 6$ while their surface area is $n\pi d^2$. Because the grains occupy the volume
fraction $1-\varphi$ of the entire rock, the total volume of the rock is $n\pi d^3 / 6(1-\varphi)$. As a
result, the specific surface area is $6(1-\varphi) / d$ .
By replacing $S$ in equation 3 with the latter expression, we find:
$$k = \frac{d^2}{72\tau^2}\frac{\varphi^3}{(1-\varphi)^2} \dots\dots\dots\dots\dots\dots\dots\dots\dots\dots\dots\dots\dots\dots\dots\dots\dots\dots \ (4)$$

which is a commonly used form of KC equation. The units used in this equation have
to be consistent. In practical use they are often not, meaning that $d$ is measured in mm
while $k$ is in mD. For these units, equation 4 can be read as:
$$k = 10^9 \frac{d^2}{72\tau^2}\frac{\varphi^3}{(1-\varphi)^2} \dots\dots\dots\dots\dots\dots\dots\dots\dots\dots\dots\dots\dots\dots\dots\dots \ (5)$$

Mavko and Nur (1997) modified this equation by introducing the percolation porosity
$\varphi_p$ below which the pore space becomes disconnected and $k$ becomes zero, although $\varphi$
is still finite:



$$k = 10^9 \frac{d^2}{72\tau^2} \frac{(\varphi - \varphi_p)^3}{(1 - \varphi + \varphi_p)^2} \dots \dots \dots \dots \dots \dots \dots \dots \dots \dots (6)$$


where, as before, $k$ is in mD, $d$ is in mm, and $\varphi$ is in fraction of one.

**Kozeny-Carman Equation with Pore Size**

As we discussed in the introduction, using the grain size in KC equation is not

consistent with the formalism where the pore space is idealized as a set of parallel
pipes.

Let us explore whether we can introduce the length parameter into KC

equation in a more logical way and reformulate it using the pore size rather than grain
size. With this goal in mind, let us recall another form of KC equation (e.g., Mavko et
al., 2009)

$$k = r^2 \frac{\varphi}{8\tau^2} = D^2 \frac{\varphi}{32\tau^2} \dots \dots \dots \dots \dots \dots \dots \dots \dots \dots \dots \dots (7)$$


where $r$ is the radius of the circular pipe that passes through the solid block and $D$ is
its diameter.

Let us assume, hence, that the porosity only depends on the size of the pipe

and is proportional to its cross-section, i.e., proportional to $D^2$. Hence, if the pore's
diameter is $D_0$ at porosity $\varphi_o$ and $D$ at porosity $\varphi$,

$$\frac{\varphi}{\varphi_0} = \frac{D^2}{D_0^2}, D^2 = D_0^2 \frac{\varphi}{\varphi_0} \dots \dots \dots \dots \dots \dots \dots \dots \dots \dots \dots \dots (8)$$


As a result, by combining Equations (7) and (8), we obtain:

$$k = D^2 \frac{\varphi}{32\tau^2} = \frac{D_0^2}{\varphi_0} \frac{\varphi^2}{32\tau^2} \dots \dots \dots \dots \dots \dots \dots \dots \dots \dots \dots (9)$$


This equation relates the permeability to porosity squared rather than cubed, the latter
as in more common forms of the KC equation. As a result, if in equation 9 we assume
$\tau$ constant, the permeability reduction due to reducing porosity will be much less
pronounced than exhibited by the Rudies data and the respective theoretical curves





will strongly overestimate the permeability data. To mitigate this effect, let us assume
that the tortuosity is not constant but rather changes with porosity.

The tortuosity is an idealized parameter that has a clear meaning within the

KC formalism but becomes fairly nebulous in a realistic pore space that is not made of
parallel cylindrical pipes. Still, numerous authors discussed the physical meaning of
tortuosity in real rock, designed experimental and theoretical methods of obtaining it,
and suggested that $\tau$ could be variable (even within the same dataset) as a function of
porosity.
Let us focus here on two tortuosity equations:
$\tau = \varphi^{-1.2}$, …………………………………………………..……………… (10)
That is derived from laboratory contaminant diffusion experiments by Boving and
Grathwohl
(2001) and
$\tau = {(1 + \varphi^{-1})}/{2}$ …………………………………………………………..……….. (11)
That is theoretically derived by Berryman (1981).

At $\varphi = 0.3$, these two equations give $\tau = 4.24$ and $2.17$, respectively. Because

KC with $\tau= 2.50$ matches the laboratory Rudies data at $\varphi = 0.3$, let us modify
equations 10 and 11 so that both produce $\tau= 2.50$ at $\varphi= 0.3$. These equations thus
modified become, respectively,
$\tau = 0.590\varphi^{-1.2}$, …………………………………………………………….. (12)
and
$\tau = 0.576(1 + \varphi^{-1})$ ………………………………………………………….. (13)

By substituting equations 12 and 13 into equation 9, we arrive at the following

two KC estimates, respectively:



$$k = 0.0898 \frac{D_0^2}{\varphi_0} \varphi^{4.4} \ldots\ldots\ldots\ldots\ldots\ldots\ldots\ldots\ldots\ldots\ldots\ldots\ldots\ldots\ldots\ldots\ldots\ldots\ldots.(14)$$


and

$$k = 0.0942 \frac{D_0^2}{\varphi_0} \frac{\varphi^4}{(1+\varphi)^2} \ldots\ldots\ldots\ldots\ldots\ldots\ldots\ldots\ldots\ldots\ldots\ldots\ldots\ldots (15)$$


with equation 14 giving the lower permeability estimate and equation 15 giving the
upper estimate for porosity below 30%. For permeability in mD and pore diameter in
mm, a multiplier $10^9$ has to be added to the right-hand sides of these equations.
Finally, by introducing the percolation porosity into these equations and using
the units mD for $k$ and mm for $D_0$, we obtain, respectively,

$$k = 0.0898 \times 10^9 \frac{D_0^2}{\varphi_0} \left(\varphi - \varphi_p\right)^{4.4} \ldots\ldots\ldots\ldots\ldots\ldots\ldots\ldots\ldots\ldots.\ldots\ldots.(16)$$


and

$$k = 0.0942 \times 10^9 \frac{D_0^2}{\varphi_0} \frac{\left(\varphi + \varphi_p\right)^4}{\left(1 + \varphi + \varphi_p\right)^2} \ldots\ldots\ldots\ldots\ldots\ldots\ldots\ldots\ldots\ldots\ldots.(17)$$



**Other Permeability-Porosity Trends and Their Explanation**
In most rocks, permeability does not follow the classic clay free trend
equations 16 and 17. The question is then how to use the KC equation to explain or
predict permeability in such formations. To address this question, we will use the KC
functional form with the grain size $d$.
Let us now recall equation 3 and modify it to be used with $k$ in mD and $S$ in
mm$^{-1}$:

$$k = \frac{10^9}{2} \frac{\varphi^3}{s^2 \tau^2} \ldots\ldots\ldots\ldots\ldots\ldots\ldots\ldots\ldots\ldots\ldots\ldots\ldots\ldots\ldots\ldots\ldots\ldots.(18)$$


Assume next that the porosity evolution is due to mixing of two distinctively
different grain sizes. The larger grain size is $d_{SS}$ while the smaller grain size is $d_{SH}$ and



$d_{SH} = \lambda d_{ss},$ ..............................................................(19)
where: $\lambda < 1$ is constant.
Let the volume fraction of the smaller grains in the rock be $C$ (we call it the
shale content). Then, by following Marion's (1990) formalism and assuming grain
mixing according to the ideal binary scheme (Figure 6), we obtain the total porosity $\varphi$
of this mixture as shown:
$\varphi = \varphi_{ss} - C(1 - \varphi_{sh})$ ...........................................(20)
for $C \leq \varphi_{ss}$, where $\varphi_{ss}$ is the porosity of the large grain framework while $\varphi_{sh}$ is that of
the small grain framework.
Recalling now the expression for the specific surface area given earlier in the
text, we obtain for the large grain framework (sand)

$$S_{ss} = \frac{6(1 - \varphi_{ss})}{d_{ss}} \quad ..............................................(21)$$


and for the shale

$$S_{sh} = \frac{6(1 - \varphi_{sh})}{d_{sh}} \quad ..............................................(22)$$


Assume next that the total specific surface area of the sand/shale mixture is the
sum of the two, the latter is weighted by the shale content:

$$S = S_{ss} + C S_{sh} = \frac{6}{d_{ss}}[1 - \varphi_{ss} + C(1 - \varphi_{ss})/\lambda] \quad ...........................(23)$$


Now, by using Equations 20 and 23 together with equation 18, we find:

$$k = \frac{10^9 \, d_{ss}^2}{72 \; \tau^2} \frac{[\varphi_{ss} - C(1 - \varphi_{sh})]^3}{[1 - \varphi_{ss} + C(1 - \varphi_{ss})/\lambda]^2} \quad .....................................(24)$$


As before, we can modify equation 24 to include the percolation porosity:

$$k = \frac{10^9 \, (\varphi - \varphi_p)^3}{2 \quad S^2\tau^2} = \frac{10^9 \, d_{ss}^2}{72 \; \tau^2} \frac{[\varphi_{ss} - C(1 - \varphi_{sh}) - \varphi_p]^3}{[1 - \varphi_{ss} + C(1 - \varphi_{ss})/\lambda]^2} \quad ...................(25)$$


where the total porosity is, as before, $\varphi = \varphi_{ss} - C(1 - \varphi_{sh})$.





## Results and Discussion

An example of using equation (6) to mimic the Rudies sandstone data (Lala, 2003) as well as the sorted Matullah sand data obtained from Belayim marine field, Gulf of Suez, Egypt is shown in Figure 1. The curve in this figure is according to Equation 6 with $d$ = 0.250 mm (for Rudies), $\tau$ = 2.5, and $\varphi_p$ = zero, 0.01, 0.02, and 0.03. The grain size in the Matullah dataset varies between 0.115 and 0.545 mm.

The Figure 2 shows the permeability normalized by the grain size squared, $d^2$. The Rudies sand data trend retains its shape. However, the Matullah sand data now form a distinct permeability-porosity trend which approximately falls on the KC theoretical curve. This fact emphasizes the effect of the grain size on the permeability in obtaining permeability-porosity trends for formations where $d$ is variable, $k$ / $d^2$ rather than $k$ alone is the appropriate argument.

Notice that although Equation 6 with $\varphi_p > 0$ mimics the permeability-porosity behavior of Rudies Formation data at high and low porosity, it somewhat underestimates the permeability in the 0.10 to 0.20 porosity range. The $\varphi_p = 0$ curve matches the data for porosity above 0.10 but overestimates the permeability in the $\varphi <$ 0.10 range. This is why in this porosity range, Bourbie et al. (1987) suggested to use a higher power of $\varphi$ (e.g., 8) instead of 3. To us, introducing a finite percolation porosity appears to be more physically meaningful. Still, no matter how we choose to alter the input parameters, it is important to remember that KC equation is based on highly idealized representations of the pore space and it is remarkable that it sometimes works (same has to be said about two other remarkable "guesses,"





Archie's law for the electrical resistivity and Raymer's equation for the P-wave
velocity, both discussed in Mavko et al., 2009).

Also, by observing the pore-space geometry evolution in Rudies sandstone,

one may conclude that the pore size is variable (Figure 3): the pores shrink with
decreasing porosity. In such a reservoir, the predicted permeability would be perfect if
we consider only the porosity (pore spaces) and grain size in prediction.

The resulting tortuosity from equations 12 & 13 plotted versus porosity in

Figure 4 rapidly increases with decreasing porosity, especially so in the porosity range
below 10%.
Let us assume that $\varphi_o = 0.30$, $D_0 = 0.10$ mm, and $\varphi_p = 0.01$. The respective curves
according to the two equations 16 & 17 are plotted on top of the Rudies and Mutallah
data in Figure 5.

The percolation porosity used here is different from 0.02 used in Equation 6.

The reason is that the current value 0.01 in Equations 16 and 17 gives a better match
to Rudies data in the lower porosity range.

Needless to say that, the concept of "pore size" is a strong idealization, same

as the concept of "grain size." We introduced it here because it is more consistent
with the KC formalism than the latter idealization. Practical reason for using the
equations with pore size is that this parameter can be inferred from the mercury
injection experiments or directly from a digital image of a rock sample.

Let us assume $d_{SS} = 0.25$ mm; $\tau = 2.5$ (fixed); and $\varphi_{ss} = \varphi_{sh} = 0.36$. The

resulting theoretical permeability estimates from equation 24 are plotted versus
porosity in Figure 6 for $\lambda = 1.00$; 0.10 ;and 0.01.

The curve for $\lambda = 0.10$ matches the Kharita Member data trend, obtained from

the Western Desert, Egypt, while that for $\lambda = 0.01$ matches the Bahariya Formation



data trend (Lala & Nahla, 2014). The curve for $\lambda = 1.00$ matches the high porosity
part of the Rudies Formation data trend.

The percolation porosity value only weakly affects the theoretical permeability

curves in the high and middle porosity ranges. This is why in Figure 6 we only show
curves with $\varphi_p = 0$.

**Conclusion**

The goal of this work is to explore permutations of the Kozeny-Carman

formalism and derive respective equations. Although the idealizations used in these
derivations are strong and sometimes lack internal consistency, the results indicate the
significant flexibility of this formalism. The variants of the KC equation shown here
can explain the various permeability-porosity trends observed in the laboratory,
sometimes within the framework of physical and geological reasoning. The predictive
ability of these equations is arguable since the input constants are not necessarily a-
priori known. Still, as in the case of bimodal mixtures, they can help with the quality
control of the existing data and forecasting of the permeability-porosity trends in
similar sedimentary textures.



**Acknowledgments**

The author is indebted to the Egyptian General Petroleum Corporation

(EGPC) for the permission to publish these laboratory results. The author is also
grateful to anonymous reviewers whose constructive comments helped to improve
this manuscript.



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

Fig.1.Porosity vs Permeability, the curves are from equation 6 with the percolation porosity (uppermost curve), 0.01, 0.02 and 0.03 (lowermost curve).





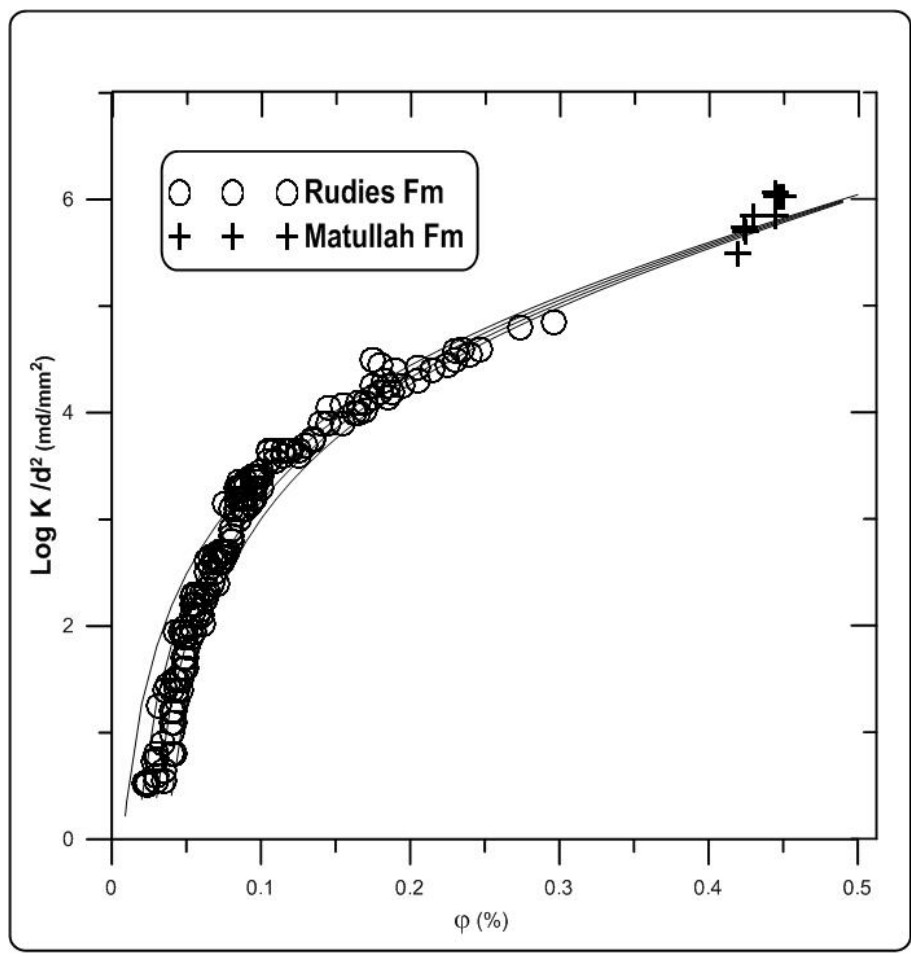

Fig.2.Porosity vs Permeability normalized by the grain size squared, the curves are from equation 6.





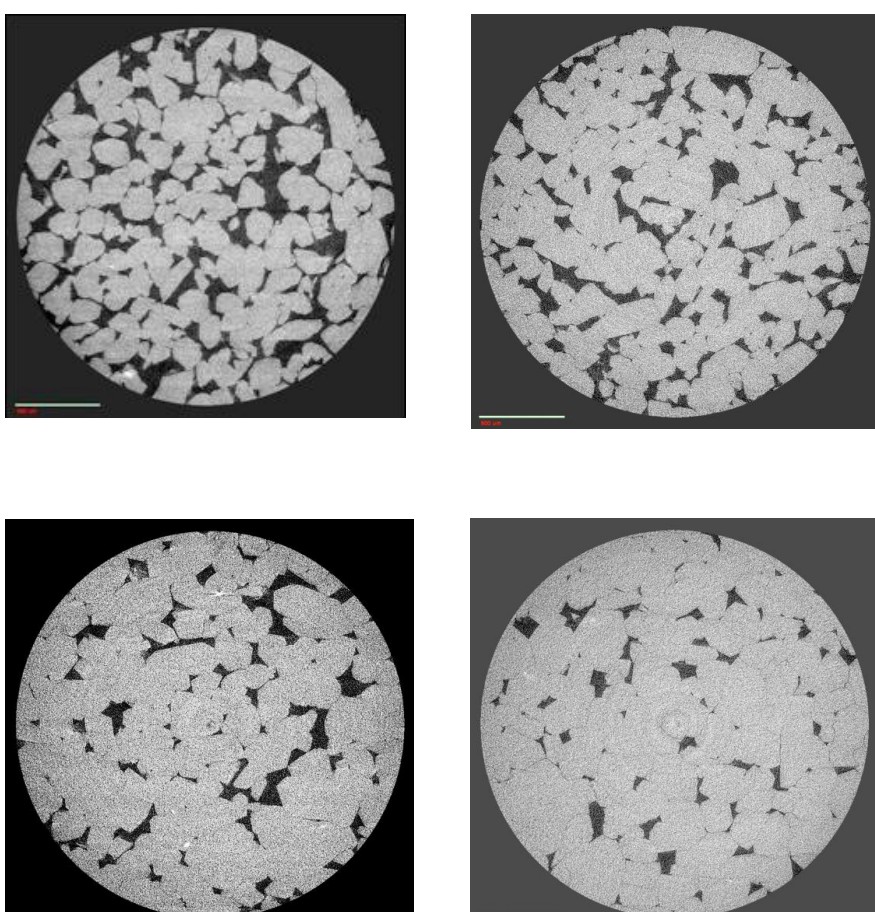

Fig.3. digital slice through four Rudies Fm samples whose porosity is gradually reducing (left to right and top to bottom). The scale bar in each image is 500µm.



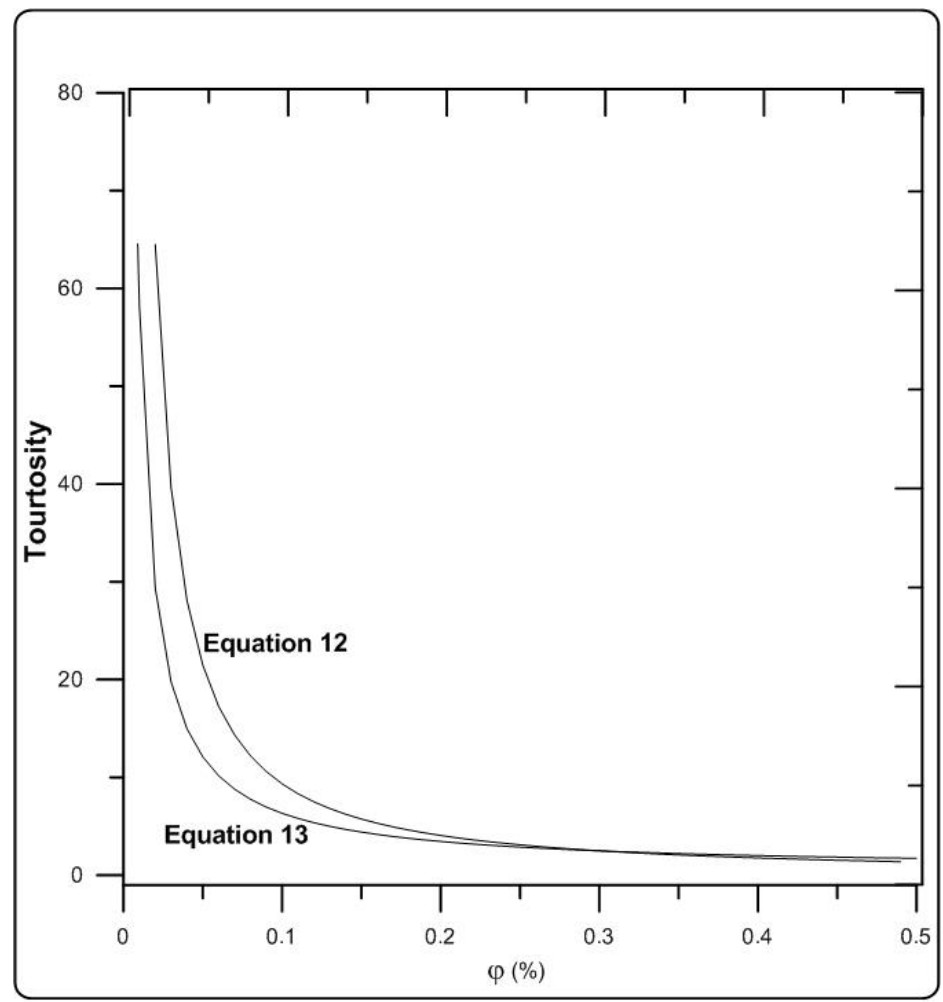

**Fig.4. Porosity versus Tourtosity.**



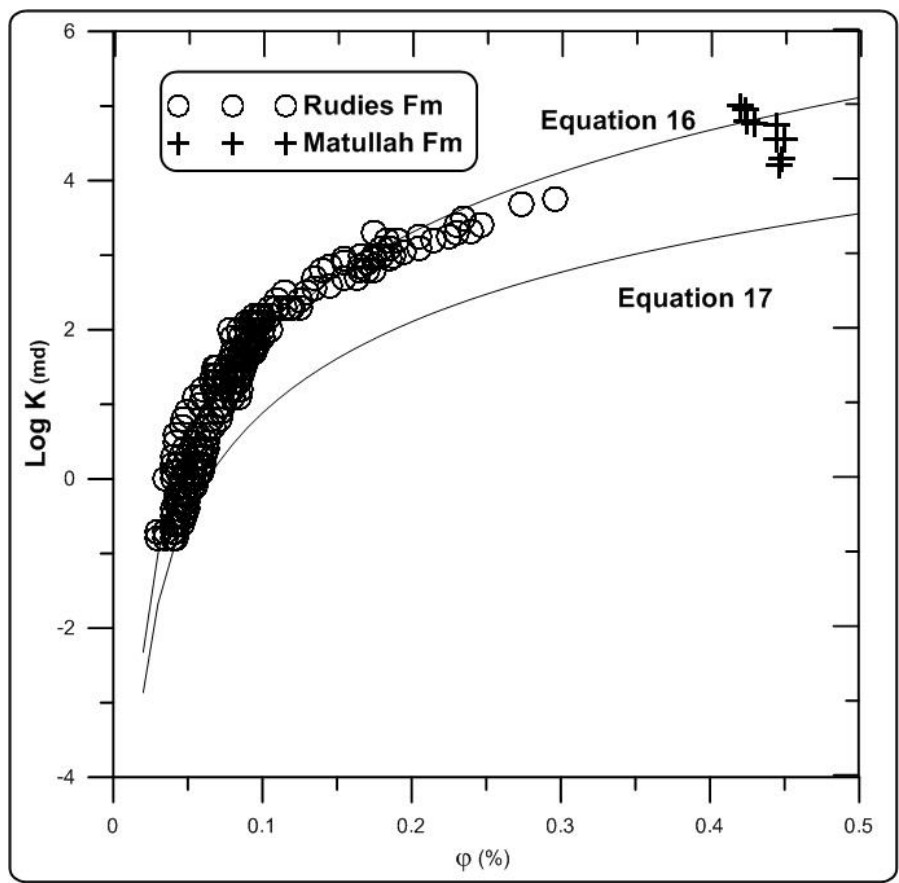

Fig.5.Porosity versus Permeability, curves are from
equation 16 and 17.



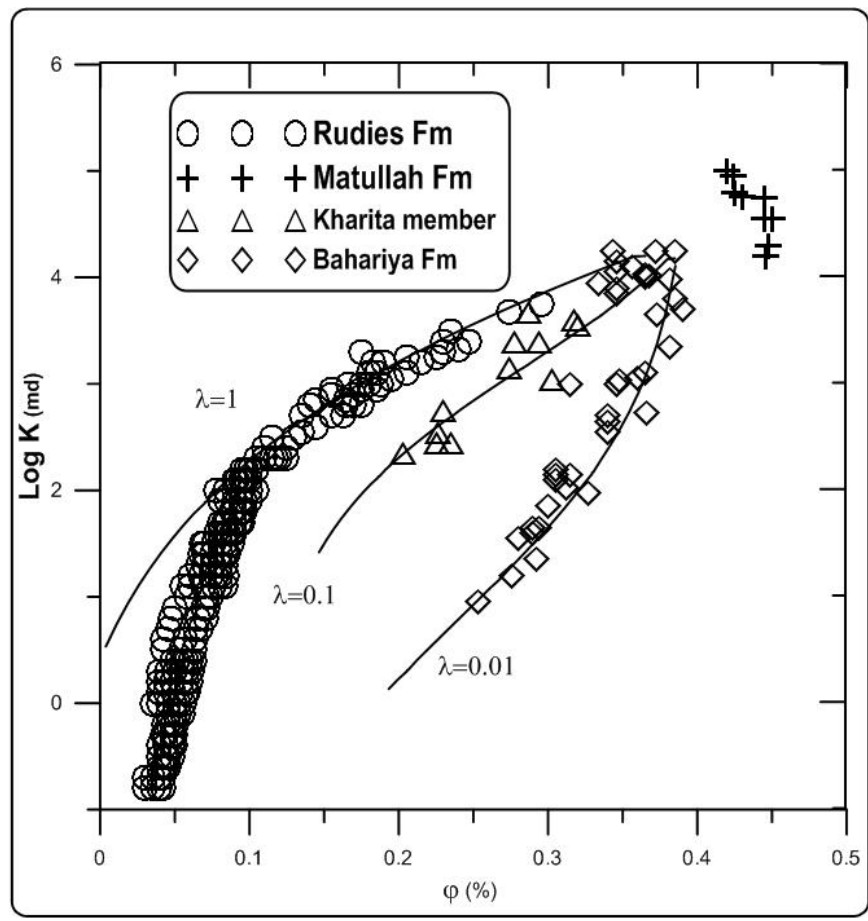

**Fig.6.Porosity vs Permeability, the curves are from equation 24.**