# Peer review of "MODIFICATIONS TO KOZENY-CARMAN MODEL To"

_Solid Earth, 2017_

## Short Comment (SC1) · 12 Mar 2017

Dear Editor, I try this new derived model on my new work and core samples raw data and it did well. I compare the calculated results obtained from the author's models with the lab petrophysical measurements obtained for my core samples. I obtain excellent match. 1- The structure of the paper is well. Abstract (explain briefly the problem, the methods used, and the results) Introduction (illustrate what is the problem, how can you solve it, the previews literature), Discussion ( explain the interpretation of the author results) the conclusions ( are well supported by the results) 2- The English is very good 3- The reference is correctly written. 4- Equations All symbols within the equations is defined and referenced 5-Figures Figure caption is under the figures. and the resolution is very good. May be authors can use colors in some figures if free. my recommendation is: The article is very good to publish in journal of solid earth.

---

## Short Comment (SC2) · 16 Mar 2017

Having read the manuscript, I find that the technique reported is certainly of interest to the Solid Earth community. So, I am pleased that this manuscript is appropriate for publication in Solid Earth for two main reasons at present. First, the manuscript as it is currently structured and presented focuses almost exclusively on technique development and reporting provisional results. I'm happy that, although this is of interest and value, this focus match the remit of Solid Earth, which is to improve our understanding of Earth surface processes. Second, both the writing style and grammar of the manuscript is very professional. Also, I am realize that all sections of the manuscript are not ambiguous and it is easy to follow the argument that the author wish to present. For this reason I recommend this work to publication in Solid Earth Journal. My opinion is I recommend that this new original paper and work is suitable for this genius journal,

because of the rather unlimited advance in understanding and analysis offered.

---

## Referee Comment (RC1) · Anonymous Referee #1 · 30 Apr 2017

Dear Dr. Rossetti,

Thank you for the opportunity to review "Modifications to Kozeny-Carman Model to Enhance Petrophyscial Relationships" by Amir Maher Sayed Lala.

The author presents a concise article that recasts the Kozeny-Carman relationship in two different ways: 1) to exploit pore size data and variable fluid path tortuosity and 2) to introduce polydisperse grain sizes into the Kozeny-Carman relationship. These models are then applied to porosity-permeability data of shaly sandstones to assess their suitability to modelling the data.

The article is generally clear and well written and the derivations of the modified Kozeny-Carman (KC) relationships are self-consistent (though there are a few errors, outlined in my major and minor comments below). However, the KC equations that

employ pore size data appear to be of little general use, since they have been empirically calibrated to fit the author's dataset, negating the purpose of the model. Further, the equations do not appear to do as good a job of fitting the author's data as the KC relationship that employs a percolation threshold (see below). In contract, the KC relationship proposed for a polydisperse grain size (the example used here is clay and sand) is more generally derived and, thus, can be applied to other datasets. However, this equation still requires the use of grain size data and so its novelty is reduced in light of the author's thesis, which purports to provide a KC relationship that is not dependent on grain size data.

Furthermore, the author discusses a dataset that has not been previously peer reviewed, without presenting it in the current manuscript. The scientific methods used to obtain these data and the data, themselves, must be provided.

The manuscript would be made clearer by a discussion that compares and contrasts the various models presented therein. I believe that this would be a useful addition to this manuscript to better highlight why the author recommends their use. I encourage the author to go further in their explanation of how these models should be applied. For example, what are the various pieces of information that are needed to adequately use the models presented: pore size, grain size, porosity?

In general, the manuscript requires significant modification to prepare it for publication. The author can do more to highlight why their work is relevant to the wider scientific community and why it is novel.

Below, I have included some major comments followed by more minor line comments. I hope that the author will find them constructive and helpful.

Major comments:

1) The derivation of the modified Kozeny-Carman relationship is confusing due to the author's choice of terminology, specifically, the use of a specific surface area term, S.

At first glance, this term does not appear to be consistent with the general definition of specific surface area in the broader literature. Specific surface area, S, is the surface area of a material per unit mass and has units of [m2/kg]. The S term used by the author in this manuscript is defined as the ratio of the total pore surface to the total sample volume (though I believe this should be total pore volume), with units of [1/m] and appears to be a reformulation of specific surface area. This term seems to be consistent with the hydraulic radius, rH, which is generally defined as the ratio between the cross-sectional area of a channel and the perimeter of the cross-section in contact with the flowing fluid. Is this true? Could the author please reference the source of this derivation of specific surface area to avoid confusion? Also, it would be useful for the general audience if the author could clarify the steps needed to transition from the more general form of specific surface area to the form used in the present manuscript.

2) In the section entitled Kozeny-Carman Equation with Pore Size, the author includes the pore diameter in the Kozeny-Carman relationship, as well as a variable tortuosity term. The author employs the work of Grathwhol (2001) and Berryman (1981) for this tortuosity term, but empirically modifies these expressions to fit their dataset. I find this problematic for several reasons. Firstly, the Kozeny-Carman relationship is, fundamentally, a model that tries to use physical parameters that can be measured in the laboratory (such as porosity, grain size/pore size, specific surface area) to predict permeability. Modifying such an expression empirically to force the relationship to fit a specific dataset is counter-productive and not of scientific interest. The parameters of the modified KC equations given in Eqs. 14 to 17 have no apparent physical meaning and, thus, do not provide meaningfully parameters to describe the measured samples, nor can these equations be applied to other datasets. Secondly, it is unclear to me why the formulation of these equations was deemed necessary by the author. As shown in Figures 1 and 2, the Kozeny-Carman relationship (modified for a percolation threshold) does a rather good job of describing the 'Rudies data' (Note: please see my comments below on the use and presentation of this data). By contrast, Eqs. 16 and 17 do a poor job of modelling this data. Further, there is no discussion of how the term Do was

chosen (or measured).

3) The author applies their derived equations to 'the Rudies data' (Line 116). There is no introduction given for the Rudies and Matullah Formations and the porosity-permeability data for these formations is not given in the manuscript. This is particularly egregious because these datasets are discussed at length in the 'Results and Discussion' section (with a cursory reference to the author's MSc thesis) but the data do not appear to have been previously published in a peer-reviewed article. If this is the case, a description of the formations and the datasets (in tabular form) should be given in this manuscript. Specifically: 1) what is the lithology and composition of the rock (this is not explicitly stated), 2) where do the samples come from, 3) why are these rocks of interest, and 4) how was porosity and permeability measured. Furthermore, where do the data for the Kharita member and Bahariya Formation (in Figure 6) come from and how were these data acquired. The source of these data must be referenced or, if the data are new, must be provided.

4) The article is currently very poorly referenced, given the extensive body of literature that has used and modified the Kozeny-Carman relationship. A conspicuous omission is the work of Costa, 2006 (GRL), which reformulates the Kozeny-Carman relationship in terms of a fractal pore space geometry. I believe this article would be of particular pertinence to the present manuscript. I have included other references that may be of use in my minor comments below.

Line comments:

Line 39: Q in Eq. 1 is the volumetric flow rate, not the volumetric flux. Volumetric flux is the volumetric flow rate divided by the cross-sectional area (i.e. Q/A) and is denoted by q. It has units of [m/s]. Please correct this in the rest of the manuscript; I have found the same mistake in lines 58, 61, and Eq. 2.

Line 47: Darcy's Law also assumes laminar fluid flow, which may be difficult to maintain in some geological materials. In many cases, fluid flow is not laminar and permeability

requires a correction for the Forchheimer and/or Klinkenberg effects.

I would further rephrase this statement to say that permeability is a fundamental rock property and remains constant, so long as the sample microstructure is unchanged – this is the reason that permeability is independent of the fluid type and the pressure conditions. Saying that permeability does not depend on fluid viscosity or pressure difference is technically correct but leads to some confusion in the text since Eq. 1 requires both these terms – should either of these terms change in Eq. 1, k inevitably changes as well. It is important to highlight the caveat that there are cases in which a high pressure difference may give rise to turbulent flow and the measured permeability is lower than the true permeability – i.e. the Forchheimer effect. Further, water and gas pore fluids interact differently with clays, yielding different values of k.

Line 48: I would avoid specifying which units are 'most commonly used' since this is not consistent across fields of study. Stick with SI units.

Line 51: Same as Line 48. Also, if the author would like to keep this sentence then please, at least, specify the 'industry'.

Line 53: Cite Kozeny, 1927; Carman, 1937; Guéguen and Palciauskas, 1994; Bernabé et al., 2010.

Line 61: Does the author mean volumetric flow rate [m3/s] (as seen in Eq. 1) instead of volume flux [m/s] (Eq. 2)?

Line 74: How is specific surface measured and why is this too difficult to do? Please specify the difficulties of measuring S in rocks to justify this statement.

Line 75: How can grain size be determined? Please give examples.

Line 80-81: "This operation is inconsistent with the KC formalism but it is useful." Please justify this comment.

Line 82: Please specify that these grains are spherical.

[Figure]

Line 82: How are the grains packed? This will influence the surface area and porosity.

Line 87: Provide a reference for this statement.

Line 116: What is the Rudies data? Up to this point there has been no discussion of it and no given reference. Further, which 'respective theoretical curves' is the author referring to?

Lines 119 to 124: Please provide references for this entire paragraph.

Line 134: Is the author referring to Eq. 9 here? Please specify the 'KC', which is being referred to. Please also reference a figure that supports this statement. Is ÏŢ=0.3 for the Rudies formation?

Lines 137 to 144: These modifications to Eqs. 10 and 11 must be justified. The author is applying an empirical correction based on a single dataset, essentially forcing a fit with no theoretical justification. These equations are, thus, only applicable to the dataset which was used to calibrate them. What is their wider application and what can we learn from them? Further, how are Do and D determined?

Lines 155-157: How are Eqs. 16 and 17 'classic clay free trends'? These are equations that have been empirically modified in the previous section of the manuscript.

Line 206: Mavko and Nur, 1997 suggest this in their article, please provide a citation.

Lines 209-211: Please remove the text in the parentheses – it is not related to the thesis of the manuscript.

Lines 214-215: What does the author mean by this statement? If the data is well defined by the sample porosity and the grain size, is there any need for the author's derivation of a KC relationship that takes into account pore diameter?

Line 223: I would argue that Eq. 17 does a very poor job of describing the Rudies data, compared to Eq. 6. The author should discuss why they think Eq. 17 does a better job of describing the data than the more general Eq. 6.

Lines 225-229: The pore size version of the KC relationship presented here appears to do a poor job of describing the Rudies data (Eq. 17, in particular) compared to the grain size derived relationship. While Eq. 16 appears to do a good job of describing the Rudies data at low porosity, it overestimates permeability at high porosity. Further, I would argue that determining grain size using image analysis is just as simple as determining pore size from image analysis.

---

## Author Comment (AC1) · 8 May 2017

Dear editor,

We all appreciate your work and the comments from reviewers, and those comments are really helpful to improve the quality of this manuscript and our related research. Now we resubmit the revised version of this MS titled: "Modifications to Kozeny-Carman Model to Enhance Petrophysical Relationships ". RESPONSE TO REFEREE REPORT(S):

1)The derivation of KC formalism is based on flow through pipe having a circular cross section with radius R. The specific surface area S (defined as the pore surface area divided by sample volume) can be expressed in terms of equation 4.

Where is the tortuosity (defined as the ratio of total flow path length to length of the sample) .

Equation 5 is exact for an ideal circular pipe geometry is presented as

A common extension of the KC relation for a circular pipe is to consider a packing of identical spheres of diameter d. Although this granular pore space geometry is not consistent with the pipe like geometry, it is common to use the original KC functional form. This allow a direct estimate of the (S) in terms of the porosity. 2) Using the grain size and model of packing of identical spheres of diameter (d) with the formalism. Explore introducing the radius of circular pipe. The parameters of modified KC equs given in 14 to 17 provide the very important parameters (pore throat radius) controlling the fluid flow in low porosity tight formation. Classical Rudies data is a special case illustrate the good description of the permeability by the grain size idealization because its clean and well sorted formation. So it will give good fit for equ 2 and 16 because $\tau$ is zero but this assumption will not valid for other medium to tight ill sorted and clayey formations. Thus equ 25 is risen. Diameter of pores is measured by capillary pressure curves. 3) table is provided Line 202 – 204 The laboratory techniques used for measuring the petrophysical parameters used in this study are presented in Lala and Nahla (2015). 4) Darcy's law is inadequate for representing high velocity fluid flow in porous media, such as near the well bore. When correlating the data for high velocity water flow through porous media, Forchheimer (1901) found that the relationship between pressure gradient an fluid velocity was no longer linear, as described by linear Darcy's flow. Forchheimer effect also known as non-Darcy effect is very important for describing additional pressure drawdown due to high fluid flow rates (Katz and Lee, 1990). Non-Darcy behavior illustrate significant effect on well performance. Non-Darcy effect play important role on effective fracture conductivity and gas well productivity. The Non Darcy flow could reduce the effective fracture conductivity and gas production and this confirmed by previous work (Guppy et al., 1982; Matias et al., Granazha et al., 2000). Traditionally, the KC equation relates the absolute permeability to porosity and grain size (d), this form is fit permeability versus porosity for data set from clean well sorted sandstone during such calculations the grain sized kept constant. One find two inconsistences in this approach, a) KC equation has been derived for a solid medium with pipe conduits rather than for a granular medium and b) even if grain size is used in this equation, it is not obvious that it doesn't vary with varying porosity. bearing this argument in mind, we explore how permeability can be predicted consistently within the KC formalism by varying the radii of the conduits. However this approach requires tourtosity evaluation during porosity reduction. Some arrive at alternate forms for the KC equation by varying tourtuosity which predict permeability and produce permeability that match measured lab data. Line 48 and 51: I specify the industry Line 47 Line 53: Done (References) Line 61: Done Line 74: Done Line 85 to 87. The specific surface area is much more difficult to measure or infer from the porosity because the granular pore spaces geometry is not consistent with the pipe like geometry model of the original K-C functional form. Line 75: Done line 87 to 91. One other parameter that can be determined in the laboratory by sieve analysis or optical microscope is the average grain size (diameter) d. The sieve analysis is the most easily understood laboratory method of determination where grains are separated on sieves of different sizes. Line 80-81: Because the KC formalism is based on cylindrical pipe model not the spherical grain packing model so this is not consistent with the KC model. However, introducing the grain size diameter improve the relationship between the permeability and the porosity so it is useful. Line 82: Done (spherical) Line 87: Done (Reference) Line 102 Line 116: Rudies Data is provided in table, Done Line 129 to 131 the Rudies Formation data obtained from Belayim marine field, Gulf of Suez, Egypt and the respective theoretical curves according to equation 6 and presented in figures 1 and 2, Line 119 to 124: Done Line 134: equ 6 , figures 1 and 2 Done line 131 Line 137 to 144: I made test for these new equations to other published data which give good results. D and Do which determined from capillary pressure curves, this different work is send also for peer review Line 155 to 157: it doesn't include the term of clay percentage $\lambda$ Line 206: citation done Line 209 to 211: Done Line 214 to 215: This is valid only for the ideal case of clean well sorted formation such as Rudies Formation, where the pore shrink with decreasing porosity. Line 223: I argue that equs 16 and 17 gives a better match than equ 6 at the lower porosity range (Tight formations). Line 225-229: The pore size concept is more consistent with the KC formalism than the grain size because it can describe permeability of tight formation at lower porosity range. Thus equations 16 and 17 give a better match at lower porosity range, also equ 16 gives a good job but overestimate permeability at high porosity but equs 6 and 24 include the grain size give poor work at lower porosity range (tight formation).

I appreciate for Editors/Reviewers' warm work earnestly, and hope that the correction will meet with approval. Once again, thank you very much for your comments and suggestions.

Please also note the supplement to this comment:
http://www.solid-earth-discuss.net/se-2017-8/se-2017-8-AC1-supplement.pdf

**Supplement:**

**Modifications to Kozeny-Carman Model to Enhance Petrophysical**

**Relationships**

Amir Maher Sayed Lala

Geophysics Department, Ain Shams University

**e-mail: amir77_lala@yahoo.com**

**Affiliation:** Geophysics Department, Fac. of Science, Ain Shams University,

Cairo, Egypt

**Abstract**

The most commonly used relationship relates permeability to porosity, grain size, and tortuosity is Kozeny-Carman formalism. When it is used to estimate the permeability behavior versus porosity, the other two parameters (the grain size and tortuosity) are usually kept constant. Here, we investigate the deficiency of the Kozeny-Carman assumption and offer alternative derived equations for the Kozeny-Carman equation, including equations where the grain size is replaced with the pore size and with varying tortuosity. We also introduced relationships for the permeability of shaly sand reservoir that answer the approximately linear permeability decreases in the log-linear permeability-porosity relationships in datasets from different locations.

**Introduction**

Darcy's law (e.g., Mavko et al., 2009) states that, the volumetric flow rate of viscous fluid Q (volume per time unit, e.g., m$^3$/s) through a sample of porous material is proportional to the cross-sectional area A and the pressure difference ΔP applied to the sample's opposite faces, and inversely proportional to the sample length L and the fluid's dynamic viscosity μ, as shown as follows:

$$Q = -k\frac{A}{\mu}\frac{\Delta P}{L} \dots\dots\dots\dots\dots\dots\dots\dots\dots\dots\dots\dots\dots\dots\dots\dots\dots\dots\dots\dots\dots\dots\dots\dots. (1)$$

The proportionality constant $k$ is called the absolute permeability. The main assumption of Darcy's law is that, $k$ does not depend on the fluid viscosity μ or pressure difference $\Delta P$ and assume a laminar fluid flow and is valid under a limited range of low velocities. All inputs in equation 1 have to be consistent units, meaning that if length is in m, pressure has to be in Pa and viscosity in Pa s. The most commonly used viscosity unit is cPs = 10$^{-3}$ Pa s. It follows from Equation 1 that the units of k are length squared, e.g., m$^2$. The most common permeability units used in the industry are Darcy (D) and/or milliDarcy (mD): $1D = 10^{-12}$ m$^2$ and 1 mD $= 10^{-15}$ m$^2$. In many cases the fluid flow is not laminar and permeability requires a correction for the Forchheimer and/or Klinkenberg effect. Forchheimer effect also known as non-Darcy effect is very important for describing additional pressure drawdown due to high fluid flow rates and could reduce the effective fracture conductivity and gas production (Guppy et al., 1982; Katz and Lee, 1990; Matins et al., 1999; Garanzha et al., 2000). Permeability is a fundamental rock property and remains constant, so long as the sample microstructure is unchanged – this is the reason that permeability is independent of the fluid type and the pressure conditions.

The Kozeny-Carman (KC) formalism (e.g., Kozeny, 1927; Carman, 1937; Guéguen and Palciauskas, 1994; Mavko et al., 2009; Bernabé et al., 2010) assumes that a porous solid can be represented as a solid block permeated by parallel cylindrical pores (pipes) whose axes may be at an angle to the direction of the pressure gradient, so that the length of an individual pipe is larger than that of the block. To relate permeability to porosity in such idealized porous solid we need to find how the volumetric flow rate $Q$ relates to the pressure gradient $\Delta P$. The solution is based on the assumption that each cylindrical pipe is circular, with radius $r$. The Navier-Stokes equations governing laminar viscous flow through a circular pipe of radius $r$ provide the following expression for the volumetric flow rate $Q$ through an individual pipe:

$$Q = -\frac{\pi r^4}{8\mu}\frac{\Delta P}{l} \,………………………………………………………………. (2)$$

where: $l$ is the length of the pipe.

Our derivation starts from the Kozeny-Carman equation by assuming that a rock includes porosity of pipe shape. The porosity, $\varphi$, and the specific surface area, $S$, can be expressed in terms of the properties of the pipe by the following relations (Mavko et al., 2009):

$$\varphi = {\pi r^2 l}/{AL} = {\pi r^2}/{A}\,\tau \;\dots\dots\dots\dots\dots\dots\dots\dots\dots\dots\dots\dots\dots\dots\dots\dots\dots\dots\dots (3)$$

Where $\tau$ is the tortuosity (defined as the ratio of total flow path length to length of the sample) .

$$S = {2\pi r l}/{AL} = {2\pi r \tau}/{A} = {2\pi r^2 \tau}/{A}\,{2}/{r} = {2\varphi}/{r} \;\dots\dots\dots\dots\dots\dots\dots\dots\dots\dots\dots\dots (4)$$

Permeability of this rock is expressed by its porosity φ and the specific surface area $S$, its length, and the number of the pipes, and using Equation 1 and 2, we get:

$$k = {\pi R^4}/{8A}\,{L}/{l} = {\pi R^4}/{8A\tau} = \frac{1}{2}\frac{\varphi^3}{S^2\tau^2} \;\dots\dots\dots\dots\dots\dots\dots\dots\dots\dots\dots\dots\dots (5)$$

where: $S$ is defined as the ratio of the total pore surface area to the total volume of the porous sample and the tortuosity $\tau$ is simply l / L , defined as the ratio of the length of the fluid path to that of the sample. Porosity can be evaluated in the laboratory or obtained from porosity logs. The specific surface area is much more difficult to measure or infer from the porosity because the granular pore spaces geometry is not consistent with the pipe like geometry model of the original K-C functional form. One other parameter that can be determined in the laboratory by sieve analysis or optical microscope is the average grain size (diameter) d. The sieve analysis is the most easily understood laboratory method of determination where grains are separated on sieves of different sizes. 
[revised manuscript text omitted]

$\qquad S_{ss} = \dfrac{6(1 - \varphi_{ss})}{d_{ss}}$ … … … … … … … … … … … … … … … … … … … … … … … … … … . . (21)

and for the shale

$\qquad S_{sh} = \dfrac{6(1 - \varphi_{sh})}{d_{\ sh}}$ … … … … … … … … … … … … … … … … … … … … … … … … … . . (22)

$\qquad$ Assume next that the total specific surface area of the sand/shale mixture is the sum of the two, the latter is weighted by the shale content:

$\qquad S = S_{ss} + C S_{sh} = \dfrac{6}{d_{ss}} [1 - \varphi_{ss} + C\,(1 - \varphi_{ss})/\lambda]$ … … … … … … … … … … … … . . . . (23)

Now, by using Equations 20 and 23 together with equation 18, we find:

$\qquad k = \dfrac{10^{9}}{72} \dfrac{d_{ss}^{2}}{\tau^{2}} \dfrac{[\varphi_{ss} - C(1 - \varphi_{sh})]^{3}}{[1 - \varphi_{ss} + C(1 - \varphi_{ss})/\lambda]^{2}}$ … … … … … … … … … … … … … … … … … … . . (24)

$\qquad$ As before, we can modify equation 24 to include the percolation porosity:

$\qquad k = \dfrac{10^{9}}{2} \dfrac{(\varphi - \varphi_{p})^{3}}{S^{2}\tau^{2}} = \dfrac{10^{9}}{72} \dfrac{d_{ss}^{2}}{\tau^{2}} \dfrac{[\varphi_{ss} - C(1 - \varphi_{sh}) - \varphi_{p}]^{3}}{[1 - \varphi_{ss} + C(1 - \varphi_{ss})/\lambda]^{2}}$ … … … … … … … … … . . . . . . (25)

where the total porosity is, as before, $\varphi = \varphi_{ss} - C(1 - \varphi_{sh})$.

**Results and Discussion**

$\qquad$ An example of using equation (6) to mimic the Rudies clean sandstone data (Lala, 2003) as well as the sorted Matullah sandstone data obtained from Belayim marine field, Gulf of Suez, Egypt is shown in Figure 1. The laboratory techniques used for measuring the petrophysical parameters used in this study are presented in Lala and

[revised manuscript text omitted]

Carman, P.C. 1937. Fluid flow through granular beds. Transactions, Institution of Chemical Engineers, London, 15: 150-166.

Katz,D. L., Lee, L. L., 1990, Natural gas engineering, New York, McGraw Hill.

Kozeny, J. 1927. Ueber kapillare Leitung des Wassers im Boden. Sitzungsber Akad. Wiss., Wien, 136 (2a): 271-306.

Garanzha, V. A., Konshin, V. N., Lyons, S. L., Papavassliou, D. V., and Qin, G., 2000, Validation of non-Darcy well models using direct numerical simulation, Chen, Ewing and Shi (eds), Numerical treatment of multiphase flow in porous media, lecture notes in physics, 552, Springer-Verlag, Berlin, 156-169.

Guppy, K. H., Cinco-ey, H. and Ramey, H. J., 1982, Pressure buildup analysis of fractured wells producing a high low rates, Journal of Petroleum Technology, 2656-2666.

Marion, D., 1990, Acoustical, mechanical and transport properties of sediments and granular materials, Ph.D. thesis, Stanford University.

Martins, J. P., Miton-Taylor, D. and Leung, H. K., 1990, The effects of non-Darcy flow in proposed hydraulic fractures, SPE 20790, proceedings of SPE Annual Technical conference, New Orleans, Louisiana, USA, Sept. 23-26.

Mavko, G., and Nur, A., 1997, The effect of a percolation threshold in the Kozeny-Carman relation, Geophysics, 62, 1480-1482.

Mavko, G., Mukerji, T., and Dvorkin, J., 2009, The rock physics handbook, Cambridge University press.

Strandenes, S., 1991, Rock physics analysis of the Brent Group Reservoir in the Oseberg  Field, Stanford Rock Physics and Borehole Geophysics Project, special volume.

Yves, Gueguen; Victor, Palciauskas 1994. Introduction to the physics of rocks. Princeton University Press.

[Figure]

Fig.1.Porosity vs Permeability, the curves are from equation 6 with the percolation porosity (uppermost curve), 0.01, 0.02 and 0.03 (lowermost curve).

[Figure]

**Fig.2.Porosity vs Permeability normalized by the grain size squared, the curves are from equation 6.**

[Figure]

Fig.3. digital slice through four Rudies Fm samples whose porosity is gradually reducing (left to right and top to bottom). The scale barin each image is 500μm.

[Figure]

**Fig.4. Porosity versus Tourtosity.**

[Figure]

Fig.5.Porosity versus Permeability, curves are from
equation 16 and 17.

[Figure]

Fig.6.Porosity vs Permeability, the curves are from equation 24.

Table (1): Porosity and Permeability of the studied samples

| No | Age | Depth (m) | Log Perm (md) | Porosity ratio | lithology |
|---|---|---|---|---|---|
| Well :113-81, Rudies Formation, Belayim land field, Gulf of Suez, Egypt | | | | | |
| 1 | | 2578.2 | -0.84 | 0.035 | sandstone |
| 2 | | 2580.25 | -0.72 | 0.035 | sandstone |
| 3 | | 2722.31 | -0.7 | 0.044 | sandstone |
| 4 | | 2800.15 | -0.56 | 0.044 | sandstone |
| 5 | | 2476.64 | -0.5 | 0.048 | sandstone |
| 6 | | 2485.72 | -0.31 | 0.05 | sandstone |
| 7 | | 2491.68 | -0.24 | 0.046 | sandstone |
| 8 | | N.A | -0.16 | 0.053 | sandstone |
| 9 | | N.A | -0.23 | 0.051 | sandstone |
| 10 | | N.A | -0.1 | 0.05 | sandstone |
| 11 | | N.A | 0.11 | 0.049 | Sandstone |
| 12 | | N.A | 0.27 | 0.048 | Sandstone |
| 13 | | N.A | 0.29 | 0.055 | Sandstone |
| 14 | | 2590.15 | 0.39 | 0.058 | Sandstone |
| 15 | Miocene | 2599.54 | 0.55 | 0.054 | Sandstone |
| 16 | | 2607.9 | 0.61 | 0.057 | Sandstone |
| 17 | | 2612.86 | 0.75 | 0.064 | Sandstone |
| 18 | | 2614 | 0.82 | 0.066 | Sandstone |
| 19 | | 2620 | 0.94 | 0.072 | Sandstone |
| 20 | | 2624.7 | 1.01 | 0.072 | Sandstone |
| 21 | | 2639 | 1.14 | 0.076 | Sandstone |
| 22 | | 2643.9 | 1.19 | 0.085 | Sandstone |
| 23 | | 2661.05 | 1.3 | 0.076 | Sandstone |
| 24 | | 2664 | 1.4 | 0.08 | Sandstone |
| 25 | | 2688.32 | 1.75 | 0.085 | Sandstone |
| 26 | | 2497.23 | 1.79 | 0.096 | Sandstone |
| 27 | | N.A. | 1.9 | 0.094 | Sandstone |
| 28 | | N.A. | 2 | 0.095 | Sandstone |
| 29 | | N.A. | 2.12 | 0.096 | Sandstone |
| 30 | | N.A. | 2.42 | 0.1 | Sandstone |
| 31 | | N.A. | 2.63 | 0.118 | Sandstone |

Table (1, cont.) : Porosity and Permeability of the studied samples

| No | Age | Depth (m) | Log Perm (md) | Porosity ratio | lithology |
|---|---|---|---|---|---|
| Well :113-81, Rudies Formation, Belayim land field, Gulf of Suez, Egypt | | | | | |
| 32 | Miocene | N.A | 2.4 | 0.125 | Sandstone |
| 33 | | N.A | 2.5 | 0.115 | Sandstone |
| 34 | | N.A | 2.55 | 0.135 | Sandstone |
| 35 | | N.A | 2.6 | 0.145 | Sandstone |
| 36 | | N.A | 2.7 | 0.155 | Sandstone |
| 37 | | N.A | 2.8 | 0.17 | Sandstone |
| 38 | | N.A | 2.85 | 0.145 | Sandstone |
| 39 | | N.A | 2.9 | 0.155 | Sandstone |
| 40 | | N.A | 2.95 | 0.185 | Sandstone |
| 41 | | N.A | 3 | 0.18 | Sandstone |
| 42 | | N.A | 3.1 | 0.18 | Sandstone |
| 43 | | N.A | 3.05 | 0.195 | Sandstone |
| 44 | | N.A | 3.2 | 0.215 | Sandstone |
| 45 | | N.A | 3.3 | 0.175 | Sandstone |
| 46 | | N.A | 3.32 | 0.24 | Sandstone |
| 47 | | N.A | 3.4 | 0.23 | Sandstone |
| 48 | | N.A | 3.5 | 0.235 | Sandstone |
| 49 | | N.A | 3.68 | 0.274 | Sandstone |
| 50 | | N.A | 3.75 | 0.296 | Sandstone |

Table (1, cont.) : Porosity and Permeability of the studied samples

| No | Age | Depth (m) | Log Perm (md) | Porosity ratio | lithology |
|---|---|---|---|---|---|
| Well :BED 1-2, Kharita member, Burg El Arab Formation, Western Desert, Egypt | | | | | |
| 207 | | N.A. | 2.45 | 0.225 | Sandstone |
| 208 | | N.A. | 2.55 | 0.226 | Sandstone |
| 209 | | N.A. | 2.45 | 0.235 | Sandstone |
| 211 | | N.A. | 2.75 | 0.23 | Sandstone |
| 212 | | N.A. | 3.15 | 0.274 | Sandstone |
| 214 | | N.A. | 3.4 | 0.277 | Sandstone |
| 217 | | N.A. | 3.4 | 0.294 | Sandstone |
| 218 | | N.A. | 3.68 | 0.287 | Sandstone |
| 220 | | N.A. | 3.05 | 0.303 | Sandstone |
| 221 | | N.A. | 3.55 | 0.32 | Sandstone |
| 222 | | N.A. | 3.6 | 0.317 | Sandstone |

Table (1, cont.) : Porosity and Permeability of the studied samples

| No | Age | Depth (m) | Log Perm (md) | Porosity ratio | lithology |
|---|---|---|---|---|---|
| Well :BED 1-2, Bahariya Formation, Western Desert, Egypt | | | | | |
| 1 | | N.A. | 0.066 | 0.18 | sandstone |
| 2 | | N.A. | 0.145 | 0.19 | sandstone |
| 3 | | N.A. | 1.22 | 0.33 | sandstone |
| 4 | | N.A. | 1.30 | 0.34 | sandstone |
| 5 | | N.A. | 0.223 | 0.2 | sandstone |
| 6 | | N.A. | 1.39 | 0.35 | sandstone |
| 7 | | N.A. | 1.56 | 0.37 | sandstone |
| 8 | | N.A. | 0.301 | 0.21 | sandstone |
| 10 | | N.A. | 0.453 | 0.23 | sandstone |
| 11 | | N.A. | 0.53 | 0.24 | sandstone |
| 13 | | N.A. | 0.68 | 0.26 | sandstone |
| 14 | Upper Cretaceous | N.A. | 0.75 | 0.27 | sandstone |
| 15 | | N.A. | 0.83 | 0.28 | sandstone |
| 16 | | N.A. | 0.97 | 0.3 | sandstone |
| 17 | | N.A. | 1.76 | 0.39 | sandstone |
| 18 | | N.A. | 1.85 | 0.4 | sandstone |
| 19 | | N.A. | 1.97 | 0.41 | sandstone |
| 20 | | N.A. | 2.1 | 0.42 | sandstone |
| 21 | | N.A. | 2.22 | 0.43 | sandstone |
| 22 | | N.A. | 1.14 | 0.32 | sandstone |
| 23 | | N.A. | 2.36 | 0.44 | sandstone |
| 24 | | N.A. | 2.52 | 0.45 | sandstone |
| 25 | | N.A. | 2.71 | 0.46 | sandstone |
| 26 | | N.A. | 2.92 | 0.47 | sandstone |
| 27 | | N.A. | 3.2 | 0.48 | sandstone |
| 28 | | N.A. | 3.57 | 0.49 | sandstone |

Table (1, cont.): Porosity and Permeability of the Studied Samples

| No | Age | Depth (m) | Log Perm (md) | Porosity ratio | lithology |
|---|---|---|---|---|---|
| Well :BM-85, Matullah Formation, Belayim marine field, Gulf of Suez, Egypt | | | | | |
| 1 | Lower senonian, upper cretaceous | 3446.03 | 4.2 | 0.446 | Sandstone |
| 2 | | 3449.03 | 4.3 | 0.448 | Sandstone |
| 3 | | 3451.14 | 4.55 | 0.445 | Sandstone |
| 5 | | 3455.17 | 4.75 | 0.445 | Sandstone |
| 7 | | 3457.44 | 4.79 | 0.425 | Sandstone |
| 9 | | 3473.45 | 4.95 | 0.424 | Sandstone |
| 10 | | 3477.23 | 5 | 0.42 | Sandstone |

Dear editor,

We all appreciate your work and the comments from reviewers, and those comments are really helpful to improve the quality of this manuscript and our related research. Now we resubmit the revised version of this MS titled:

"**Modifications to Kozeny-Carman Model to Enhance Petrophysical Relationships** ".

RESPONSE TO REFEREE REPORT(S):

1)The derivation of KC formalism is based on flow through pipe having a circular cross section with radius R. The specific surface area S (defined as the pore surface area divided by sample volume) can be expressed in terms of equation 4.

$$\varphi = {\pi r^2 l}/{AL} = {\pi r^2}/{A}\,\tau \ldots\ldots\ldots\ldots\ldots\ldots\ldots\ldots\ldots\ldots\ldots\ldots\ldots\ldots\ldots\ldots\ldots\ldots\ldots\ldots\ldots\ldots. (3)$$

Where $\tau$ is the tortuosity (defined as the ratio of total flow path length to length of the sample) .

$$S = {2\pi r l}/{AL} = {2\pi r \tau}/{A} = {2\pi r^2 \tau}/{A}\,{2}/{r} = {2\varphi}/{r} \ldots\ldots\ldots\ldots\ldots\ldots\ldots\ldots.\ldots\ldots\ldots\ldots.. (4)$$

Equation 5 is exact for an ideal circular pipe geometry is presented as

$$k = {\pi R^4}/{8A}\,{L}/{l} = {\pi R^4}/{8A\tau} = \frac{1}{2}\frac{\varphi^3}{S^2\tau^2} \ldots\ldots\ldots\ldots\ldots\ldots\ldots\ldots\ldots\ldots\ldots\ldots\ldots\ldots\ldots\ldots (5)$$

A common extension of the KC relation for a circular pipe is to consider a packing of identical spheres of diameter d. Although this granular pore space geometry is not consistent with the pipe like geometry, it is common to use the original KC functional form. This allow a direct estimate of the (S) in terms of the porosity.

2) Using the grain size and model of packing of identical spheres of diameter (d) with the formalism. Explore introducing the radius of circular pipe. The parameters of modified KC equs given in 14 to 17 provide the very important parameters (pore throat radius) controlling the fluid flow in low porosity tight formation. Classical Rudies data is a special case illustrate the good description of the permeability by the grain size idealization because its clean and well sorted formation. So it will give good fit for equ 2 and 16 because $\tau$ is zero but this assumption will not valid for other medium to tight ill sorted and clayey formations. Thus equ 25 is risen. Diameter of pores is measured by capillary pressure curves.

3) table is provided

Line 202 – 204 The laboratory techniques used for measuring the petrophysical parameters used in this study are presented in Lala and Nahla (2015).

4) Darcy's law is inadequate for representing high velocity fluid flow in porous media, such as near the well bore. When correlating the data for high velocity water flow through porous media, Forchheimer (1901) found that the relationship between pressure gradient an fluid velocity was no longer linear, as described by linear Darcy's flow. Forchheimer effect also known as non-Darcy effect is very important for describing additional pressure drawdown due to high fluid flow rates (Katz and Lee, 1990). Non-Darcy behavior illustrate significant effect on well performance. Non-Darcy effect play important role on effective fracture conductivity and gas well productivity. The Non Darcy flow could reduce the effective fracture conductivity and gas production and this confirmed by previous work (Guppy et al., 1982; Matias et al., Granazha et al., 2000).

Traditionally, the KC equation relates the absolute permeability to porosity and grain size (d), this form is fit permeability versus porosity for data set from clean well sorted sandstone during such calculations the grain sized kept constant. One find two inconsistences in this approach, a) KC equation has been derived for a solid medium with pipe conduits rather than for a granular medium and b) even if grain size is used in this equation, it is not obvious that it doesn't vary with varying porosity. bearing this argument in mind, we explore how permeability can be predicted consistently within the KC formalism by varying the radii of the conduits. However this approach requires tourtosity evaluation during porosity reduction. Some arrive at alternate forms for the KC equation by varying tourtuosity which predict permeability and produce permeability that match measured lab data.

Line 48 and 51: I specify the industry Line 47

Line 53: Done (References)

Line 61: Done

Line 74: Done Line 85 to 87. The specific surface area is much more difficult to measure or infer from the porosity because the granular pore spaces geometry is not consistent with the pipe like geometry model of the original K-C functional form.

Line 75: Done line 87 to 91. One other parameter that can be determined in the laboratory by sieve analysis or optical microscope is the average grain size (diameter) d. The sieve analysis is the most easily understood laboratory method of determination where grains are separated on sieves of different sizes.

Line 80-81: Because the KC formalism is based on cylindrical pipe model not the spherical grain packing model so this is not consistent with the KC model. However, introducing the grain size diameter improve the relationship between the permeability and the porosity so it is useful.

Line 82: Done (spherical)

Line 87: Done (Reference) Line 102

Line 116: Rudies Data is provided in table, Done Line 129 to 131 the Rudies Formation data obtained from Belayim marine field, Gulf of Suez, Egypt and the respective theoretical curves according to equation 6 and presented in figures 1 and 2,

Line 119 to 124: Done

Line 134: equ 6 , figures 1 and 2 Done line 131

Line 137 to 144: I made test for these new equations to other published data which give good results. D and Do which determined from capillary pressure curves, this different work is send also for peer review

Line 155 to 157: it doesn't include the term of clay percentage λ

Line 206: citation done

Line 209 to 211: Done

Line 214 to 215: This is valid only for the ideal case of clean well sorted formation such as Rudies Formation, where the pore shrink with decreasing porosity.

Line 223: I argue that equs 16 and 17 gives a better match than equ 6 at the lower porosity range (Tight formations).

Line 225-229: The pore size concept is more consistent with the KC formalism than the grain size because it can describe permeability of tight formation at lower porosity range. Thus equations 16 and 17 give a better match at lower porosity range, also equ 16 gives a good job but overestimate permeability at high porosity but equs 6 and 24 include the grain size give poor work at lower porosity range (tight formation).

I appreciate for Editors/Reviewers' warm work earnestly, and hope that the correction will meet with approval. Once again, thank you very much for your comments and suggestions.

---

## Short Comment (SC3) · 9 May 2017

**COMMENTS ON SOLID EARTH SE-2017-8**

**by Paul Glover**

**Comment**

The author should be congratulated on an interesting paper which promises to be very useful. I have a few points to make, which I hope will improve it further.

1. The Kozeny-Carman relationship is an interesting place to start considering that it suffers the considerable deficiency that it cannot take account of the effect of isolated and dead-end pores on rock permeability. Walker and Glover (2010) discusses this problem in detail, while comparing the KC approach with a number of other approaches. This paper should be cited in any revision.

2. It could be argued that a brief cited discussion of alternative base models (listed in Walker and Glover (2010)) should be inserted into the first part of the paper such that the reader knows that other and better models than the KC model now exist, especially the various versions of the RGPZ model (Rashid et al., 2015a;b; Glover et al., 2006). Nevertheless, I do not think that the choice of the KC model invalidates the paper – it is true that this model is still widely used despite its failings, and any modification that attempts to improve it should be welcomed.

3. The author compares the model against a number of datasets. A table of the main properties of each of these datasets should be inserted before the results such that the reader knows its main properties (mean and standard deviation porosities, grain sizes and grain sizes).

4. The results and discussion of results is fairly meagre and could be expanded.

5. The new model contains 8 parameters. The large number of parameters goes a long was to making the fits to data much easier to achieve. Consequently, it is really important to use the new equation only when all or most of the input parameters are known – if you vary too many parameters until the best fit is observed it is possible to obtain a fit with unrealistic parameters. I feel that this danger should be discussed towards the end of the paper.

6. As this model contains 8 parameters, I think that it would be reasonable to carry out a sensitivity analysis in a section of its own (just before the Results section). In this section there should be a figure or figures that show how the resulting permeability varies as each of the parameters is varied between physically reasonable limits, while the other parameters are held constant at a reasonable value. This section will help the reader understand how the equation works as well as testing the equation's limiting values. The section should also discuss which, if any, parameters have less effect on the resulting permeabilities, and which parameters affect the permeabilities most.

**References**

GLOVER, P.W.J., ZADJALI, I.I. and FREW, K.A., 2006. Permeability prediction from MICP and NMR data using an electrokinetic approach. Geophysics, 71(4).

GLOVER, P.W.J. and WALKER, E., 2009. Grain-size to effective pore-size transformation derived from electrokinetic theory. Geophysics, 74(1).

WALKER, E. and GLOVER, P.W.J., 2010. Permeability models of porous media: Characteristic length scales, scaling constants and time-dependent electrokinetic coupling. Geophysics, 75(6), pp. E235-E246.

RASHID, F., GLOVER, P.W.J., LORINCZI, P., COLLIER, R. and LAWRENCE, J., 2015a. Porosity and permeability of tight carbonate reservoir rocks in the north of Iraq. Journal of Petroleum Science and Engineering, 133, pp. 147-161.

RASHID, F., GLOVER, P.W.J., LORINCZI, P., HUSSEIN, D., COLLIER, R. and LAWRENCE, J., 2015b. Permeability prediction in tight carbonate rocks using capillary pressure measurements. Marine and Petroleum Geology, 68, pp. 536-550.

**Paul Glover**

---

## Referee Comment (RC2) · Anonymous Referee #2 · 12 May 2017

Please see attachd document.

Please also note the supplement to this comment:
http://www.solid-earth-discuss.net/se-2017-8/se-2017-8-RC2-supplement.pdf
* * *
[Figure]

This is a review of the paper "Modifications to Kozeny–Carman model to enhance petrophysical relationships" by Amir M. S. Lala. It presents an interesting study where a modification of the famous Kozeny-Carmon equation is given to estimate the absolute permeability from porosity, tortuosity and grain or pore size distributions. Below are my comments and suggestions to improve this paper.

**General Comments:**

1. One major criticism is that the paper needs to include recent studies in the topic. There are many important and relevant papers that have investigated the KC model and proposed alternative forms. For this, the author needs to bring out clearly what is new in his work.  For instance, Nooruddin and Hossain, 2011 did modify the tortuosity in the KC model making it a function of porosity and other parameters. The author here needs to explain his new idea very clearly and distinguish it from previous work.

**Specific Comments:**

- Eqs 2, 3, 4: citations are needed here.
- In Eq.2 : change q to Q
- Line 72: use the mathematical symbol used in Eq.2 to clearly indicate the definition of tortuosity – it looks as L^-1, while you mean (\ell/L)
- Line 91 – 93: This depends on how you define porosity in the KC model in which it is most likely nothing but the effective porosity which - by definition - accounts for connected pores only (see for instance Nooruddin and Hossain, 2011).  However, you

**Supplement:**

This is a review of the paper "Modifications to Kozeny–Carman model to enhance petrophysical relationships" by Amir M. S. Lala. It presents an interesting study where a modification of the famous Kozeny-Carmon equation is given to estimate the absolute permeability from porosity, tortuosity and grain or pore size distributions. Below are my comments and suggestions to improve this paper.

**General Comments:**

1. One major criticism is that the paper needs to include recent studies in the topic. There are many important and relevant papers that have investigated the KC model and proposed alternative forms. For this, the author needs to bring out clearly what is new in his work. For instance, Nooruddin and Hossain, 2011 did modify the tortuosity in the KC model making it a function of porosity and other parameters. The author here needs to explain his new idea very clearly and distinguish it from previous work.

**Specific Comments:**

- Eqs 2, 3, 4: citations are needed here.
- In Eq.2 : change q to Q
- Line 72: use the mathematical symbol used in Eq.2 to clearly indicate the definition of tortuosity – it looks as L^-1, while you mean ($\ell/L$)
- Line 91 – 93: This depends on how you define porosity in the KC model in which it is most likely nothing but the effective porosity which - by definition - accounts for connected pores only (see for instance Nooruddin and Hossain, 2011). However, you define porosity in the KC model as total porosity, including isolated pores, which I don't think is correct, since isolated pores do not contribute to the permeability of the sample.
- Line 118: The idea that tortuosity changes with porosity is not new; other researchers have addressed this point specifically (e.g., Wyllie and Rose, 1950; Winsauer et al., 1952). Other researchers (e.g., Nooruddin and Hossain, 2011) have modified the KC model by specifically modifying the tortuosity term to include the impact of porosity. Please be clear in distinguishing your work from previous studies and show clearly your new contributions.
- Eqs. 10 and 11: indicate why you choose these models over other tortuosity models in the literature.
- Line 134: you mentioned Rudies data but did not give any description of it. I recommend having a separate section on the description of this dataset, especially if it has not been published before, showing main geological features, and including statistical measures. If the dataset has been published, then you need to cite that paper.
- Line-190: from where did you get model's parameters; did you use curve fitting?
- Line – 197: What d value did you use in the normalization? is it a constant value or a distribution? And if it is a distribution, from where did you get it?

- Line-206: As I mentioned previously that the effective porosity should be used in the KC model to be consistent with its derivations which explicitly accounts for connected pores only.
- Line193 – remove "the" before Figure 2.
- Finally, in addition to the above comments, I encourage the author to consider the comments made by Paul Glover in his se-2017-8-SC3-suplement.pdf document. All his comments are valuable; especially comments # 3,4,5, and 6.

---

## Author Comment (AC2) · 16 May 2017

RESPONSE TO REFEREE REPORT(S):

Eqs 2, 3, 4: citations are needed here. Done the following expression for the volumetric flow rate Q through an individual pipe (Faber 1995):

. . . . . . . . . .. . . . . .. . . . . . . .. . . . . . .. . . . . . . .. . . . . . . .. . . . . . . .. . . . . . . .. . . . . .. . . . .. . . . (2) can be expressed in terms of the properties of the pipe by the following relations (Mavko et al., 2009): ðÍŚ§2 . . . . . . .. . . . . . .. . . . . . . .. . . . . . . .. . . . . . . . .. . . . . . .. . . . . . . .. . . . . . . .. . . . (3) Where ï£¡ is the tortuosity (defined as the ratio of total flow path length to length of the

In Eq.2 : change q to Q Done 28 Line 72: use the mathematical symbol used in Eq.2 to clearly indicate the definition of tortuosity – it looks as Lˆ-1, while you mean (\ell/L)

Done

Line 77 Where ï£¡ is the tortuosity (defined as the ratio of total flow path length (ï£¡) to length of the sample (ï£¡)) .

Line 91 – 93: This depends on how you define porosity in the KC model in which it is most likely nothing but the effective porosity which - by definition - accounts for connected pores only (see for instance Nooruddin and Hossain, 2011). However, you define porosity in the KC model as total porosity, including isolated pores, which I don't think is correct, since isolated pores do not contribute to the permeability of the sample. Yes sir, The porosity in the original form of the K-C model is the total porosity so I follow the Mavko and Nur 1997 to introduce the term of the percolation porosity.

Line 118: The idea that tortuosity changes with porosity is not new; other researchers have addressed this point specifically (e.g., Wyllie and Rose, 1950; Winsauer et al., 1952). Other researchers (e.g., Nooruddin and Hossain, 2011) have modified the KC model by specifically modifying the tortuosity term to include the impact of porosity. Please be clear in distinguishing your work from previous studies and show clearly your new contributions. The new of my work is that both equation 16 and 17 which I can use to describe the permeability of tight formations at lower porosity range. Eqs 16 and 17 give a best fit at the lower porosity range (tight formations) 29

Line-190: from where did you get model's parameters; did you use curve fitting? All the model parameters included in the equation 24 by the mathematical derivation and success after that in measured permeability description as shown in figure 6

Line 223: I argue that eqs 16 and 17 give a better match than eq 6 in the lower porosity range. Eqs. 10 and 11: indicate why you choose these models over other tortuosity models in the literature. Because the first one is derived from laboratory experiment and the second from the theoretical and for me I am trust of both models too much.

Line 134: you mentioned Rudies data but did not give any description of it. I recommend having a separate section on the description of this dataset, especially if it has not been published before, showing main geological features, and including statistical measures. If the dataset has been published, then you need to cite that paper. Done I provide the table

Line – 197: What d value did you use in the normalization? is it a constant value or a distribution? And if it is a distribution, from where did you get it with grain diameter d = 0.250 mm is the best representative value for Rudies formation obtained from the sieve and microscopic analysis.

30 I appreciate for Editors/Reviewers' warm work earnestly, and hope that the correction will meet with approval. Once again, thank you very much for your comments and suggestions.

Please also note the supplement to this comment:
http://www.solid-earth-discuss.net/se-2017-8/se-2017-8-AC2-supplement.pdf

[Figure]

**Supplement:**

| 1  | Modifications to Kozeny-Carman Model to Enhance Petrophysical              |
|----|----------------------------------------------------------------------------|
| 2  | Relationships                                                              |
| 3  | Amir Maher Sayed Lala                                                      |
| 4  | Geophysics Department, Ain Shams University                                |
| 5  | e-mail: amir77_lala@yahoo.com                                       |
| 6  | Affiliation: Geophysics Department, Fac. of Science, Ain Shams University, |
| 7  | Cairo, Egypt                                                               |
| 8  |                                                                            |
| 9  |                                                                            |
| 10 |                                                                            |
| 11 |                                                                            |
| 12 |                                                                            |
| 13 |                                                                            |
| 14 |                                                                            |
| 15 |                                                                            |
| 16 |                                                                            |
| 17 |                                                                            |
| 18 |                                                                            |
| 19 |                                                                            |
| 20 |                                                                            |
| 21 |                                                                            |
|    |                                                                            |
|    |                                                                            |

**Abstract**

| 24 | The most commonly used relationship relates permeability to porosity, grain                                                                                 |
|----|-------------------------------------------------------------------------------------------------------------------------------------------------------------|
| 25 | size, and tortuosity is Kozeny-Carman formalism. When it is used to estimate the                                                                            |
| 26 | permeability behavior versus porosity, the other two parameters (the grain size and                                                                         |
| 27 | tortuosity) are usually kept constant. Here, we investigate the deficiency of the Kozeny-                                                                   |
| 28 | Carman assumption and offer alternative derived equations for the Kozeny-Carman                                                                             |
| 29 | equation, including equations where the grain size is replaced with the pore size and                                                                       |
| 30 | with varying tortuosity. We also introduced relationships for the permeability of shaly                                                                     |
| 31 | sand reservoir that answer the approximately linear permeability decreases in the log-                                                                      |
| 32 | linear permeability-porosity relationships in datasets from different locations.                                                                            |
| 33 |                                                                                                                                                             |
| 34 | Introduction                                                                                                                                                |
| 35 | Darcy's law (e.g., Mavko et al., 2009) states that, the volumetric flow rate of                                                                             |
| 36 | viscous fluid Q (volume per time unit, e.g., m 3 /s) through a sample of porous material                                                         |
| 37 | is proportional to the cross-sectional area A and the pressure difference $\Delta P$ applied to                                                             |
| 38 | the sample's opposite faces, and inversely proportional to the sample length L and the                                                                      |
| 39 | fluid's dynamic viscosity $\mu$ , as shown as follows:                                                                                                      |
| 40 | $Q = -k \frac{A}{\mu} \frac{\Delta P}{L} \dots \dots$ |
| 41 | The proportionality constant $k$ is called the absolute permeability. The main                                                                              |
| 42 | assumption of Darcy's law is that, k does not depend on the fluid viscosity $\mu$ or pressure                                                               |
| 43 | difference $\Delta P$ and assume a laminar fluid flow and is valid under a limited range of low                                                             |
| 44 | velocities. All inputs in equation 1 have to be consistent units, meaning that if length is                                                                 |
| 45 | in m, pressure has to be in Pa and viscosity in Pa s. The most commonly used viscosity                                                                      |
| 46 | unit is $cPs = 10^{-3}$ Pa s. It follows from Equation 1 that the units of k are length squared,                                                            |
| 47 | e.g., m 2 . The most common permeability units used in the industry are Darcy (D) and/or                                                         |

milliDarcy (mD):  $1D = 10^{-12} \text{ m}^2$  and  $1 \text{ mD} = 10^{-15} \text{ m}^2$ . In many cases the fluid flow is 48 49 not laminar and permeability requires a correction for the Forchheimer and/or 50 Klinkenberg effect. Forchheimer effect also known as non-Darcy effect is very 51 important for describing additional pressure drawdown due to high fluid flow rates and 52 could reduce the effective fracture conductivity and gas production (Guppy et al., 1982; 53 Katz and Lee, 1990; Matins et al., 1999; Garanzha et al., 2000). Permeability is a 54 fundamental rock property and remains constant, so long as the sample microstructure 55 is unchanged – this is the reason that permeability is independent of the fluid type and 56 the pressure conditions.

57 The Kozeny-Carman (KC) formalism (e.g., Kozeny, 1927; Carman, 1937; 58 Guéguen and Palciauskas, 1994; Mavko et al., 2009; Bernabé et al., 2010) assumes that 59 a porous solid can be represented as a solid block permeated by parallel cylindrical 60 pores (pipes) whose axes may be at an angle to the direction of the pressure gradient, 61 so that the length of an individual pipe is larger than that of the block. To relate 62 permeability to porosity in such idealized porous solid we need to find how the 63 volumetric flow rate Q relates to the pressure gradient  $\Delta P$ . The solution is based on the 64 assumption that each cylindrical pipe is circular, with radius r. The Navier-Stokes 65 equations governing laminar viscous flow through a circular pipe of radius r provide 66 the following expression for the volumetric flow rate Q through an individual pipe 67 (Faber 1995):

68

70

71 where: l is the length of the pipe.

72 Our derivation starts from the Kozeny-Carman equation by assuming that a rock includes porosity of pipe shape. The porosity,  $\varphi$ , and the specific surface area, S, can 73 74 be expressed in terms of the properties of the pipe by the following relations (Mavko et 75 al., 2009):

- 76
- Where  $\tau$  is the tortuosity (defined as the ratio of total flow path length (l) to length of 77 78 the sample (L)).

80 Permeability of this rock is expressed by its porosity  $\varphi$  and the specific surface 81 area S, its length, and the number of the pipes, and using Equation 1 and 2, we get:

83 where: S is defined as the ratio of the total pore surface area to the total volume of the 84 porous sample and the tortuosity  $\tau$  is simply 1/L, defined as the ratio of the length of 85 the fluid path to that of the sample. Porosity can be evaluated in the laboratory or 86 obtained from porosity logs. The specific surface area is much more difficult to measure 87 or infer from the porosity because the granular pore spaces geometry is not consistent 88 with the pipe like geometry model of the original K-C functional form. One other 89 parameter that can be determined in the laboratory by sieve analysis or optical 90 microscope is the average grain size (diameter) d. The sieve analysis is the most easily 91 understood laboratory method of determination where grains are separated on sieves of 92 different sizes. 
[revised manuscript text omitted]
 laboratory techniques used for measuring the petrophysical parameters used in this study are presented in Lala and Nahla (2015). The curve in this figure is according to Equation 6 with d = 0.250 mm (for Rudies),  $\tau = 2.5$ , and  $\varphi_p =$  zero, 0.01, 0.02, and 0.03. The grain size in the Matullah dataset varies between 0.115 and 0.545 mm.

209

Figure 2 shows the permeability normalized by the grain size squared,  $d^2$ . The Rudies sand data trend retains its shape. However, the Matullah sand data now form a distinct permeability-porosity trend which approximately falls on the KC theoretical curve. This fact emphasizes the effect of the grain size on the permeability in obtaining permeability-porosity trends for formations where *d* is variable,  $k/d^2$  rather than *k* alone is the appropriate argument.

216

217 Notice that although Equation 6 with  $\varphi_p > 0$  mimics the permeability-porosity 218 behavior of Rudies Formation data at high and low porosity, it somewhat 219 underestimates the permeability in the 0.10 to 0.20 porosity range. The  $\varphi_p = 0$  curve 220 matches the data for porosity above 0.10 but overestimates the permeability in the  $\varphi

Fig.1.Porosity vs Permeability, the curves are from equation 6 with the percolation porosity (uppermost curve), 0.01, 0.02 and 0.03 (lowermost curve).